# In situ grown oxygen-vacancy-rich copper oxide nanosheets on a copper foam electrode afford the selective oxidation of alcohols to value-added chemicals

Mustafa Khan [1], Asima Hameed[1], Abdus Samad [2], Talifhani Mushiana[1], Muhammad Imran Abdullah [3✉], Asma Akhtar[1], Raja Shahid Ashraf[3], Ning Zhang[4], Bruno G. Pollet[5], Udo Schwingenschlögl [2✉] & Mingming Ma [1✉]

Selective oxidation of low-molecular-weight aliphatic alcohols like methanol and ethanol into carboxylates in acid/base hybrid electrolytic cells offers reduced process operating costs for the generation of fuels and value-added chemicals, which is environmentally and economically more desirable than their full oxidation to $CO_2$. Herein, we report the in-situ fabrication of oxygen-vacancies-rich CuO nanosheets on a copper foam (CF) via a simple ultrasonication-assisted acid-etching method. The CuO/CF monolith electrode enables efficient and selective electrooxidation of ethanol and methanol into value-added acetate and formate with ~100% selectivity. First principles calculations reveal that oxygen vacancies in CuO nanosheets efficiently regulate the surface chemistry and electronic structure, provide abundant active sites, and enhance charge transfer that facilitates the adsorption of reactant molecules on the catalyst surface. The as-prepared CuO/CF monolith electrode shows excellent stability for alcohol oxidation at current densities >200 mA·cm$^2$ for 24 h. Moreover, the abundant oxygen vacancies significantly enhance the intrinsic indicators of the catalyst in terms of specific activity and outstanding turnover frequencies of 5.8k s$^{-1}$ and 6k s$^{-1}$ for acetate and formate normalized by their respective faradaic efficiencies at an applied potential of 1.82 V vs. RHE.

[1] Hefei National Laboratory for Physical Sciences at the Microscale, Department of Chemistry, University of Science and Technology of China, Hefei, Anhui 230026, China. [2] Physical Science and Engineering Division, King Abdullah University of Science and Technology (KAUST), Thuwal 23955-6900, Saudi Arabia. [3] Department of Chemistry, Government College University Lahore, Lahore 54000, Pakistan. [4] School of Biology, Food and Environment, Hefei University, Hefei, Anhui 230022, China. [5] Pollet Research Group, Hydrogen Research Institute (HRI), Université du Québec à Trois-Rivières, 3351 Boulevard des Forges, Trois-Rivières, QC G9A 5H7, Canada. ✉email: mimran@mail.ustc.edu.cn; udo.schwingenschlogl@kaust.edu.sa; mma@ustc.edu.cn

The electrochemical oxidation of low-molecular-weight aliphatic alcohols like methanol and ethanol are promising half-cell anodic reaction reactions that can sufficiently replace the energy intensive oxygen evolution reaction (OER) taking place in various acid/base hybrid electrolytic cells[1,2]. Alcohol oxidation reaction (AOR) at the anode is the key half-reaction that determines the overall efficiency of an alkaline electrolyzer. Currently, AOR is mostly carried out with expensive Pt and their alloys with other noble metals like Pt-Ru and Pt-Pd, etc[3–5]. The noble-metal based catalysts completely oxidize the alcohols into $CO_2$ and the generated CO intermediate will poison the catalyst, therefore the high price, as well as vulnerable nature of these catalysts towards CO poisoning restrict their use for AOR[6,7]. On the other hand low-cost earth abundant transition metal-based catalysts selectively oxidizes alcohols into carboxylates as value-added chemicals. The latter path is economically and environmentally advantageous in terms of lowering the overall cost of the process as well as avoiding $CO_2$ emissions into the environment. For instance, selective oxidation of methanol provides formate as a highly useful intermediate involved in various industrial applications like printing process, fabric dying, pesticides, pharmaceuticals, and as a liquid fuel in the direct formic acid fuel cells. Currently, the industrial-scale production of formate/formic acid involves the combination of CO with methanol at 80 °C and 40 atm followed by the subsequent hydrolysis of the resulting methyl formate, which is an energy and cost-intensive process. This results in a fourfold higher price per metric ton of formate compared to methanol[8–10]. Similarly, acetic acid/acetate is also a very important commodity chemical and finds a wide range of industrial applications[11]. Hence, the electrochemical synthesis of formate and acetate from low-cost methanol and ethanol under ambient conditions is a promising and economical alternative way. However, most of the reported catalysts for AOR have not shown good selectivity towards carboxylate in alkaline media, and their product selectivity in terms of turnover frequency (TOF) and Faradaic efficiencies are rarely reported.

First-row transition metal-based (Ni, Co, and Cu) electrocatalysts including their oxides and hydroxides have been considered promising alternatives to noble metal-based electrocatalysts for AOR[1,12–15]. Among them, the oxide and hydroxides of Cu have been reported to perform efficiently in different applications like, electrochemical energy conversions, photocatalytic $H_2$ production, dye-sensitized solar cells, sensing devices, lithium-ion batteries, etc[12,16–19]. Easy and fast fabrication of an efficient catalyst is the bottom line for the successful commercialization of any specific application. Therefore, researchers are focusing on faster and simple methods for the fabrication of an efficient catalyst. Recently Ananthaj et al has reported a faster anodization method that takes only 80 s to fabricate a dense array of $Cu(OH)_2$-CuO nanoneedles on Cu foil substrate via reasonably lower constant potential in 1 M KOH[20]. He further modifies the anodization method by introducing Cu foam substrate containing Ni impurities to grow Cu–O/(Ni) nanowires, which takes 10 min[21]. The Ni impurity in Cu–O/(Ni) nanowires significantly enhances the methanol oxidation reaction (MOR) activity compared to the $Cu(OH)_2$-CuO nanoneedles arrays. The catalysts prepared with the anodization method developed by Ananthaj et al take sufficiently lower time and have better MOR performance compared to the Cu-based catalysts reported earlier by electrochemical as well as other physical and chemical methods[12,22,23]. We have prepared in-situ CuO nanostructure on copper foam (CF) via a simple and mild chemical oxidation method. Unlike other chemical oxidation methods which usually take longer time and require different chemical oxidants under strict reaction conditions, our method uses $O_2$ in the air as an oxidant under ambient conditions. Our developed method also

takes only 10 min for the fabrication of CuO nanostructure like that of the reported faster anodization method discussed earlier. Moreover, the developed chemical oxidation method also imparts some defects in the form of oxygen vacancies ($V_O$) into the CuO nanostructure which are known to improve the catalytic performances of the electrocatalyst[24–29]. As a result, the obtained CuO catalyst show superior MOR and EOR performance specifically in term of intrinsic properties like specific activity and exceptional TOF for value-added formate and acetate compared to the Cu-based oxides/hydroxides as well as most of the oxides/hydroxides reported for other transition metals. Furthermore, most of the AOR catalysts are nanomaterials that are loaded on conductive support with the help of binding materials. During long-term AOR at high current densities, the electrical and mechanical contact between conductive support and active catalyst could be easily damaged, which marks a severe decay in the catalytic performance of the electrode[24,30]. Therefore, the direct growth of the active catalysts on a conductive substrate to form a monolith electrode is vital for an enhanced AOR performance[31,32].

Herein, we prepare $V_O$-rich CuO nanosheets on CF via a simple ultrasonication-assisted acid-etched method (Fig. 1). Ultrasound energy greatly enhances the acid etching by cavitation effect and oxidizes the surface of CF at room temperature without the addition of any other oxidants. The cavitation process in acidic conditions largely stimulates the formation of nanopits, which act as a nucleation site for the precipitation of $Cu_2O$ throughout the air-drying step. During AOR, the $Cu_2O$/CF converts into CuO/CF nanosheets accompanied by a relatively higher abundance of $V_O$. Benefitting from the combined effect of $V_O$-rich nanosheet structure having higher surface area, the obtained CuO/CF monolith electrode exhibits an outstanding electrocatalytic activity and stability for the selective electro-oxidation of methanol and ethanol into value-added formate and acetate with higher selectivity and Faraday efficiencies.

## Results and Discussion

### Characterization of the as-prepared CuO/CF.
As shown in Fig. 1, the self-supported anodic catalyst namely CuO/CF was synthesized by simple ultrasonication of bare CF (B-CF) at room temperature in 2 M HCl solution. Fig. 2a, b, and Supplementary Fig. 2a, b represent the scanning electron microscopic (SEM) images of the B-CF and ultrasonically assisted acid-etched CF (UAAE-CF), which clearly shows the smooth surface of B-CF develops nanopits after ultrasonication in 2 M HCl solution followed by the subsequent washing and air-drying step. The morphology of UAAE-CF evolved during the stability test in an

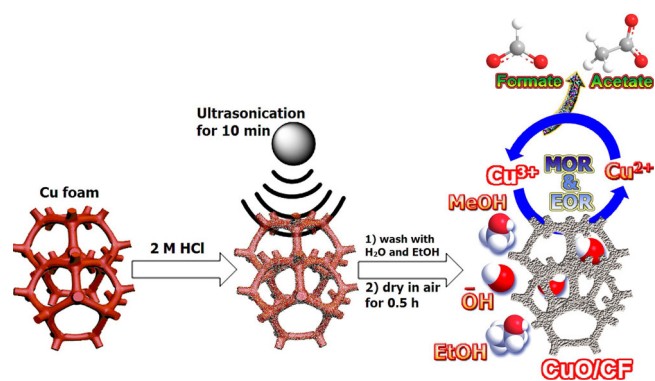

**Fig. 1 Preparation and application of CuO nanosheets on Cu foam.** Schematic illustration of the CuO/CF structure preparation via ultrasonication and its application towards selective oxidation of alcohols into value-added products.

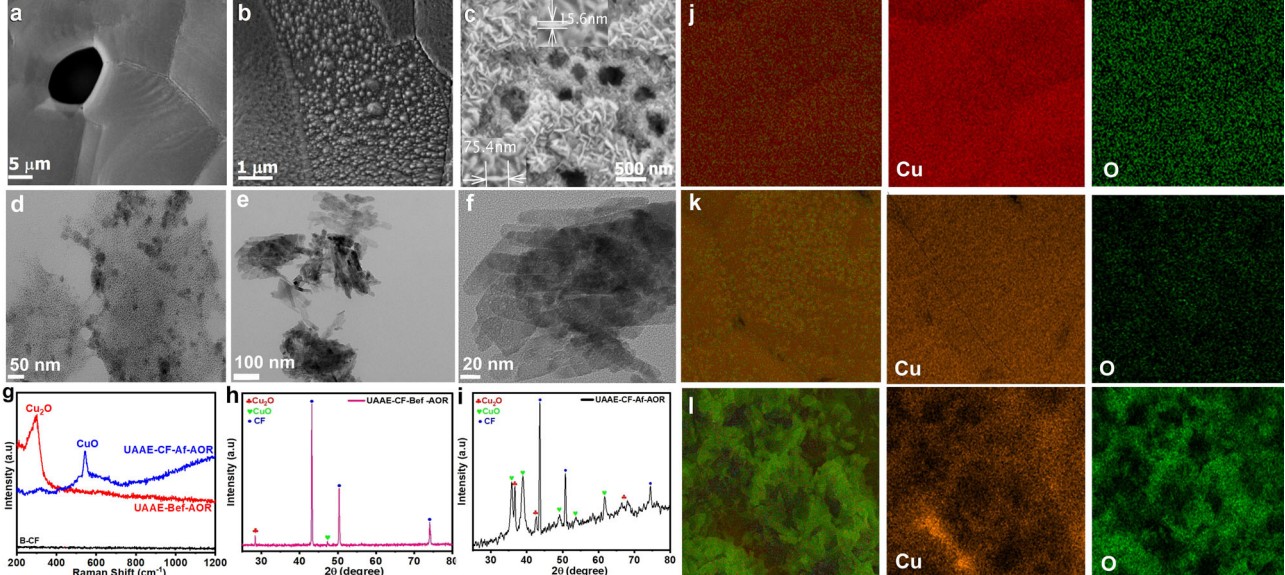

**Fig. 2 SEM images of B-CF and UAAE-CF. a** B-CF. **b** UAAE-CF before the AOR stability test. **c** UAAE-CF after the AOR stability test. TEM images of. **d** UAAE-CF before the AOR stability test. **e**, **f** UAAE-CF after the AOR stability test. **g** Raman spectra of B-CF and UAAE-CF before and after the AOR stability test. **h** XRD patterns UAAE-CF. **i** XRD patterns of UAAE-CF after the AOR stability test. Elemental mapping of. **j** B-CF. **k** UAAE-CF. **l** UAAE-CF after the AOR stability test.

alkaline solution containing 1 M alcohol (methanol or ethanol) and formed nanosheets. Furthermore, the obtained thickness and length of nanosheets were measured to be 15.6 nm and 75.4 nm respectively as shown in Fig. 2c. Moreover, transmission electron microscopy (TEM) further confirms the formation of nanoparticles and nanosheets on the surface of UAAE-CF and UAAE-CF after long-term stability test respectively (Fig. 2d-f). Figure 2j-l displays the elemental mapping for B-CF, UAAE-CF, and UAAE-CF after long-term stability test. As shown Cu and O are homogeneously distributed over the synthesized nanoparticles and nanosheets.

Raman spectroscopy depicted in Fig. 2g of the studied samples reveals that B-CF displayed a smooth Raman spectrum. However, UAAE-CF exhibits a strong Raman shift at 300 cm$^{-1}$, which corresponds to the Cu–O stretching mode of $Cu_2O$. Moreover, a new pronounced peak arises around 542 cm$^{-1}$ in UAAE-CF after the AOR stability tests which can be attributed to the Cu–O stretching band of CuO as presented in Fig. 2g[12]. Besides, the XRD finding of UAAE-CF discloses the polycrystalline nature of the material. The peak at 28.5° is associated with the in-situ deposited layer of $Cu_2O$ in UAAE-CF, while the smaller peak around 47.5° is due to the formation of CuO (Fig. 2h). However, during the AOR stability test new and more intense peaks related to CuO originate in UAAE-CF, as shown in Fig. 2i. Hence, both XRD and Raman analyses confirm that the nanosheets formed during the AOR stability test are mostly comprised of CuO.

The surface composition, chemical state, and defects in UAAE-CF before and after long-term AOR were examined with X-ray photoelectron spectroscopy. A broader XPS survey of C 1s, Cu 2p, and O 1s of UAAE-CF before and after long-term AOR is displayed in Supplementary Fig. 3 which is exclusively composed of Cu and O without any detected impurity. The Cu 2p peaks at 932.63 eV and 952.55 eV are assigned to $Cu^{1+}$ in $Cu_2O$, while the shoulder peak at 934.45 eV along with a small satellite peak around 945 eV confirms the presence of $Cu^{2+}$ oxidation state in UAAE-CF before the AOR stability test (Fig. 3a)[33]. The broad O 1s peak presented in Fig. 3d is deconvoluted into three typical O peaks at binding energies of ~530.3 eV, 531.7 eV, and 533 eV (denoted as O1, O2, and O3) representing lattice oxygen ($O_L$) in

$Cu_2O$, $V_O$ and absorbed water molecule on the surface of $Cu_2O$ respectively, while the relatively large area of $V_O$ peak at 531.7 eV confirms the existence of $V_O$ in UAAE-CF[1,34,35]. Generally, $V_O$ rapidly depletes at higher oxidizing potential under alkaline conditions which severely deteriorates the catalyst activity during the long-term stability test[36,37]. The quantitative analysis of $V_O$ in UAAE-CF catalyst before and after the long-term AOR test was carried out by taking the ratio of $V_O/O_L$[19]. Interestingly in our case, the ratio of $V_O/O_L$ increases from 2.26 to 3.51 after the long-term AOR at extremely harsh oxidizing conditions and thus favors a significant increase in the relative concentration of $V_O$ during AOR (Fig. 3e). Moreover, the Cu 2p peak presented in Fig. 3a-c reveals that the concentration of $Cu^{2+}$ largely increases from 27.5 to 73% after long-term AOR with a negative shift of 0.5 eV. As shown in Fig. 3d-f the O 1s peak also shifts towards negative binding energy after long-term AOR, the peak at 529.6 eV and 531.3 eV belongs to $O_L$ and $V_O$ in CuO respectively. The above-mentioned negative shift in binding energies of Cu 2p and O 1s peaks indicates that long-term AOR significantly enhances the formation of new $V_O$ on the catalyst surface[19,38]. As shown, the UAAE-CF catalyst evolves during long-term catalysis and converts mainly to the CuO which is the main active catalyst ingredient for AOR, so the final catalyst will be regarded as (as-prepared CuO/CF).

**Electrochemical characterization of the as-prepared CuO/CF.** First, the as-prepared CuO/CF self-supported electrode was electrochemically characterized in the $N_2$-purged 1 M KOH solution. As presented in the Supplementary Fig. 7, the CV curve exhibits a well-defined redox couple in the region between 1.26 and 1.42 V vs. RHE, related to the $Cu^{2+}/Cu^{3+}$ redox couple. According to the literature, $Cu^{3+}$ specie (formed in the presence of $OH^-$ ion) is the main active site for the electrooxidation of organic molecules like methanol and glucose[12,19]. $V_O$ are considered to enhance the adsorption of $OH^-$ ion on the catalyst surface which further facilitates the formation of $Cu^{3+}$ species. The concentration of redox species (I*) on the catalyst surface was estimated from their respective CV curves collected at different scan rates, see Supplementary Fig. 8a-f. The estimated surface coverage

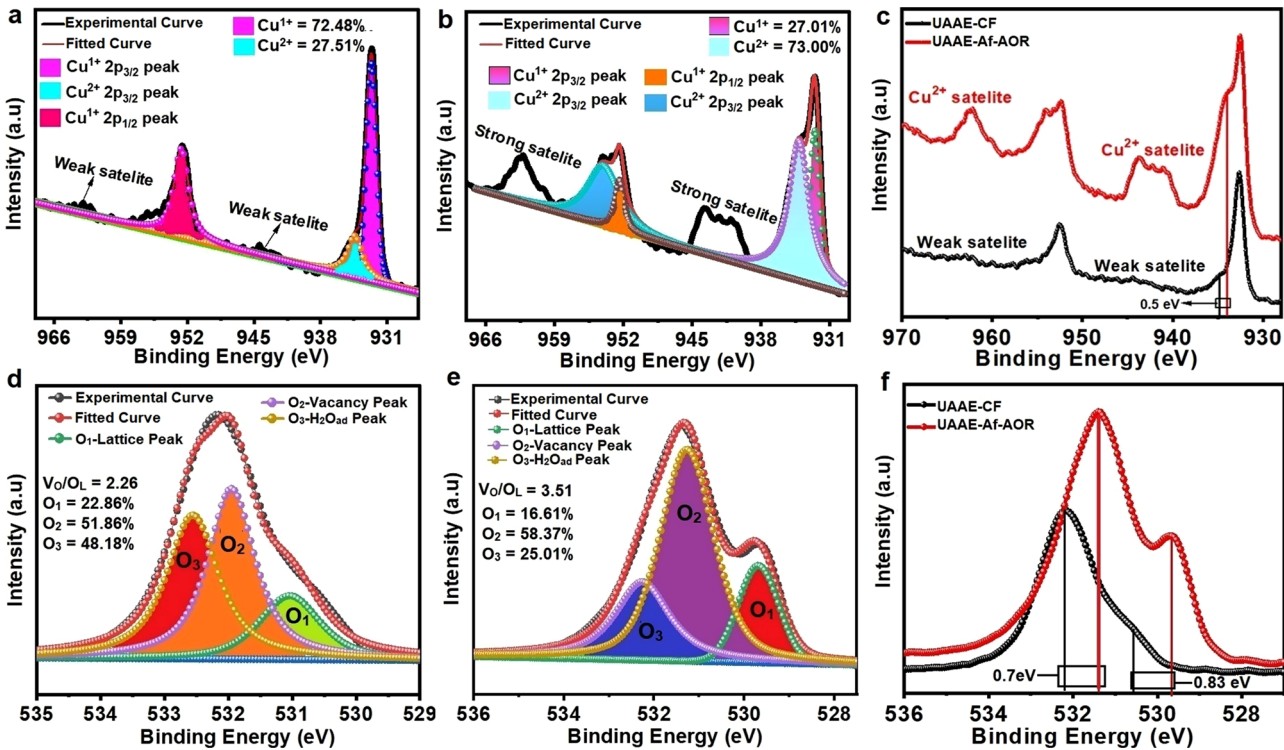

**Fig. 3 Core level of XPS spectrum of UAAE-CF. a** Cu 2p spectrum of UAAE-CF before AOR stability test. **b** Cu 2p spectrum of UAAE-CF after AOR stability test. **c** Comparison of Cu 2p spectra of UAAE-CF before and after AOR stability test, **d** O 1s spectrum of UAAE-CF before AOR stability test. **e** O 1s spectrum of UAAE-CF after AOR stability test. **f** Comparison of O 1s spectra of UAA-CF before and after the AOR stability test.

$I^* = 1.43 \times 10^{-7}$ mol·cm$^{-2}$ for the as-prepared CuO/CF was much higher than that of B-CF ($5.44 \times 10^{-8}$ mol·cm$^{-2}$) and other reported non-noble metal-based electrocatalysts as shown in Supplementary Table 1-3. Furthermore, linear proportionality between the anodic peak current density and square root of the potential sweep rate reveals that Cu$^{2+}$/Cu$^{3+}$ redox couple takes place via proton diffusion process depicted in Supplementary Fig. 8c, f. Whereas the obtained proton diffusion coefficient ($D_{H+}$) for the as-prepared CuO/CF ($2.17 \times 10^{-8}$ cm$^2$·s$^{-1}$) is also higher than the B-CF ($2.61 \times 10^{-9}$ cm$^2$·s$^{-1}$) and other reported non-noble metal-based AOR catalysts see Supplementary Note 3 and Supplementary Table 1 for further details.

**AOR performance of the as-prepared CuO/CF.** The anodic AOR performance of the as-prepared CuO/CF was evaluated in N$_2$-purged 1 M KOH containing 1 M alcohols (1 M EtOH or EtOH). As presented in Fig. 4a, the voltammograms collected in the absence and presence of 1 M EtOH or MeOH in 1 M KOH are quite different. In the presence of 1 M EtOH or MeOH, a sudden rise in current densities was observed beyond 1.34 V vs. RHE. Alongside the activity of the CuO/CF electrode was also examined with SCV to eliminate the possible exaggerated activity measured with potentiodynamic polarization CV curves which may come from the double layer charging current (Fig. 4b and Supplementary Fig. 5a-f)[39]. Supplementary Fig. 6a-f, display the comparative AOR activity obtained for CuO/CF with LSV/SCV techniques. A negligible difference in AOR activity was observed for CuO/CF measured with both techniques in the potential window ranging from 1 to 1.85 V vs. RHE. Meanwhile going onward from 1.85 V vs. RHE towards higher potential the difference in AOR activity measured with LSV/SCV increases steadily, where a percentage difference of 8.2% and 13.9% were noticed for MOR and EOR at maximum applied potential of 2.023 V vs. RHE see Supplementary Fig. 6c, f. The relatively

higher consumption of methanol and ethanol at higher potential may also contribute to the noticeable difference in MOR and EOR activity measured with LSV/SCV at higher applied potential. The obtained onset potential (1.34 V vs. RHE) for AOR in this study over the as-prepared CuO/CF is reasonably lower than the B-CF and other recently reported non-noble metal-based AOR catalysts as presented in Supplementary Fig. 9a, b, and Supplementary Table 2, 3. Hence AOR onset oxidation takes place earlier than the onset oxidation potential of Cu$^{2+}$/Cu$^{3+}$ redox couple in the as-prepared CuO/CF and B-CF (Supplementary Fig. 7). However, when the applied potential becomes more positive during forward CV scan in AOR, a sudden and more pronounced increase in oxidation current density is experienced after 1.42 V vs. RHE. Remarkably, the anodic potential in AOR for the as-prepared CuO/CF is reduced by at least 250 mV at current densities of 50, 100, 150, and 200 mA·cm$^{-2}$ as compared to OER (Fig. 4c and Supplementary Note 4).

Given the outstanding AOR performance of the as-prepared CuO/CF electrode, we further analyze the underlying sources of such an enhanced electrocatalytic activity. The reaction kinetics involved in AOR were investigated with Tafel plots derived from their respective SCV curves with 100% iR drop correction[40]. Fig. 4e demonstrates that the OER on the as-prepared CuO/CF experiences a sluggish reaction kinetic and contributes to a large Tafel slope of (125 mV·dec$^{-1}$). However, a dramatic decrease in the Tafel slopes was evidenced after the addition of 1 M MeOH (111 mV·dec$^{-1}$) and 1 M EtOH (101 mV·dec$^{-1}$), which undoubtedly reflects the much faster kinetics of MOR and EOR compared to the OER. Charge transfer kinetics for AOR on the as-prepared CuO/CF and B-CF were also examined via electrochemical impedance spectroscopy (Supplementary Fig. 10a, b and Supplementary Fig. 11a, b). As displayed in Supplementary Fig. 11a, b, in the presence of only 1 M KOH the as-prepared CuO/CF acts like a capacitive material and the straight line

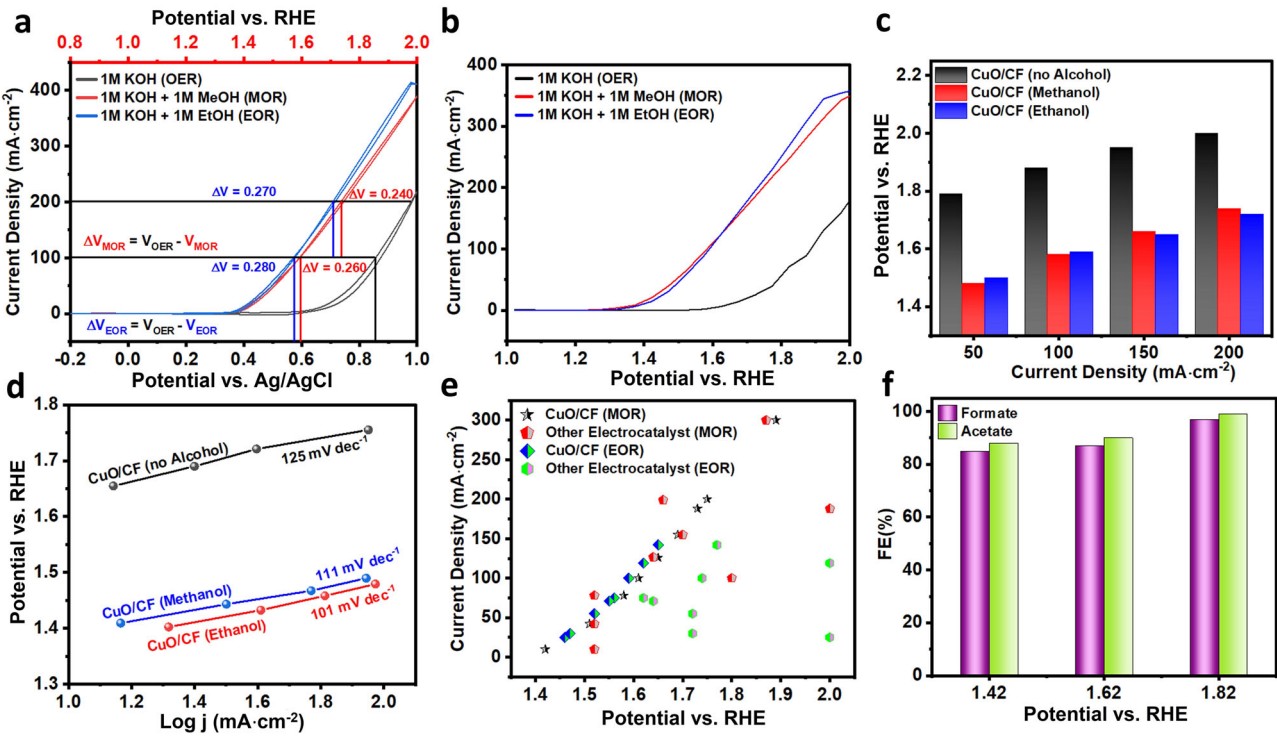

**Fig. 4 AOR performance of the as-prepared CuO/CF. a** CV curves collected over the as-prepared CuO/CF in $N_2$-purged 1 M KOH with and without 1 M ethanol or methanol. **b** SCV curves collected over the as-prepared CuO/CF in $N_2$-purged 1 M KOH with and without 1 M ethanol or methanol. **c** Summary of the anodic potentials at 50, 100, 150, and 200 mA·cm$^{-2}$ for the as-prepared CuO/CF in 1 M KOH with and without 1 M ethanol (EtOH) or methanol (MeOH) based on their respective SCV data. **d** Comparison of EOR, MOR, and OER Tafel plots obtained from their respective SCVs collected over the as-prepared CuO/CF with 100% iR correction. **e** Comparison with recently reported non-noble metal-based MOR and EOR catalysts, star data represent activity of the as-prepared CuO/CF (see Supplementary Table 2–3 for more details). **f** Summary of Faraday's efficiencies for acetate and formate.

(Warburg impedance) tends towards the *y*-axis of the Nyquist plot. However, in the presence of 1 M MeOH or 1 M EtOH, the as-prepared CuO/CF exhibits depressed semicircles associated with low charge transfer resistance ($R_{CT}$) of 1.95 Ω or 1.65 Ω, respectively. These results suggest that MOR and EOR on the surface of the as-prepared CuO/CF are charge-transfer controlled reactions and governed by the kinetics of the reaction. Furthermore, these results were also validated by taking the CV curves for MOR and EOR with different scan rates. As shown, increasing scan rate has no effect on the MOR and EOR current densities, which manifests the electrooxidation of both methanol and ethanol over the as-prepared CuO/CF are regulated by the kinetics of the reaction (Supplementary Fig. 14c, d and Supplementary Fig. 16a, b). Moreover, increasing the concentration of alcohols from 0.1 to 2 M, the anodic current density increases linearly for both EOR and MOR, which substantiates that the selective oxidation of alcohols to value-added products takes place more easily at sufficiently lower energy input compared to OER (see Supplementary Note 5, Supplementary Fig. 12a-f, and Supplementary Fig. 13a, b). Fig. 4e demonstrates the outstanding electrocatalytic EOR and MOR performance of the as-prepared CuO/CF compared to the recently reported non-noble metal-based electrocatalyst for EOR and MOR, see Supplementary Table 2, 3 for the detailed literature.

To further analyze the underline reason behind such an enhanced EOR and MOR activity for the as-prepared CuO/CF. Specific activities for both B-CF and as-prepared CuO/CF electrodes were calculated and compared. The number of accessible Cu sites (ECAS) for the B-CF and as-prepared CuO/CF was calculated from the charge integration peaks of backward EOR and MOR CV sweeps collected at a scan rate of 200 mV·s$^{-1}$ as shown in Supplementary Fig. 14e-h and Supplementary Note 6.

The specific activities for B-CF and as-prepared CuO/CF were obtained by normalizing the EOR and MOR LSVs by relative ECAS as presented in Supplementary Fig. 15a, b. As presented the calculated specific activity trend for B-CF and as-prepared CuO/CF matches well with EOR and MOR apparent activity trend, however, the difference in the number of ECAS for B-CF and as-prepared CuO/CF is not so pronounced, we attribute the enhanced specific activity in the case of as-prepared CuO/CF due to the presence of abundant $V_O$ as is evident from the density functional theory calculations. From these results, it is evident that the presence of abundant $V_O$, as well as a relatively higher ECAS, enhances the EOR and MOR activities intrinsically for as-prepared CuO/CF monolith electrode which make them superior Cu-based electrocatalyst for alkaline EOR and MOR.

**AOR product characterization**. The oxidation products of EOR and MOR at different applied potentials were quantitatively analyzed via nuclear magnetic resonance (NMR) spectroscopy see Supplementary Note 1 and Supplementary Fig. 18a, b for further details. As shown in Supplementary Fig. 19a-d, only acetate and formate were identified as the main oxidation products by $^{13}$CNMR and $^1$HNMR after 3 h of EOR and MOR at different potentials. Fig. 4f displays the calculated Faraday efficiency for acetate ($FE_{acetate}$) and formate ($FE_{formate}$), as shown the as-prepared CuO/CF exhibits a high $FE_{acetate}$ and $FE_{formate}$ (>85%) in a broad range of applied potential. A maximum $FE_{acetate}$ (99%) and $FE_{formate}$ (97%) were achieved at 1.82 V vs. RHE. TOF is another important intrinsic indicator that determines the efficiency of a catalyst for the target reaction. However, before reporting TOF, one must ensure accurate determination of TOF free from any misleading calculations see Supplementary Note 7

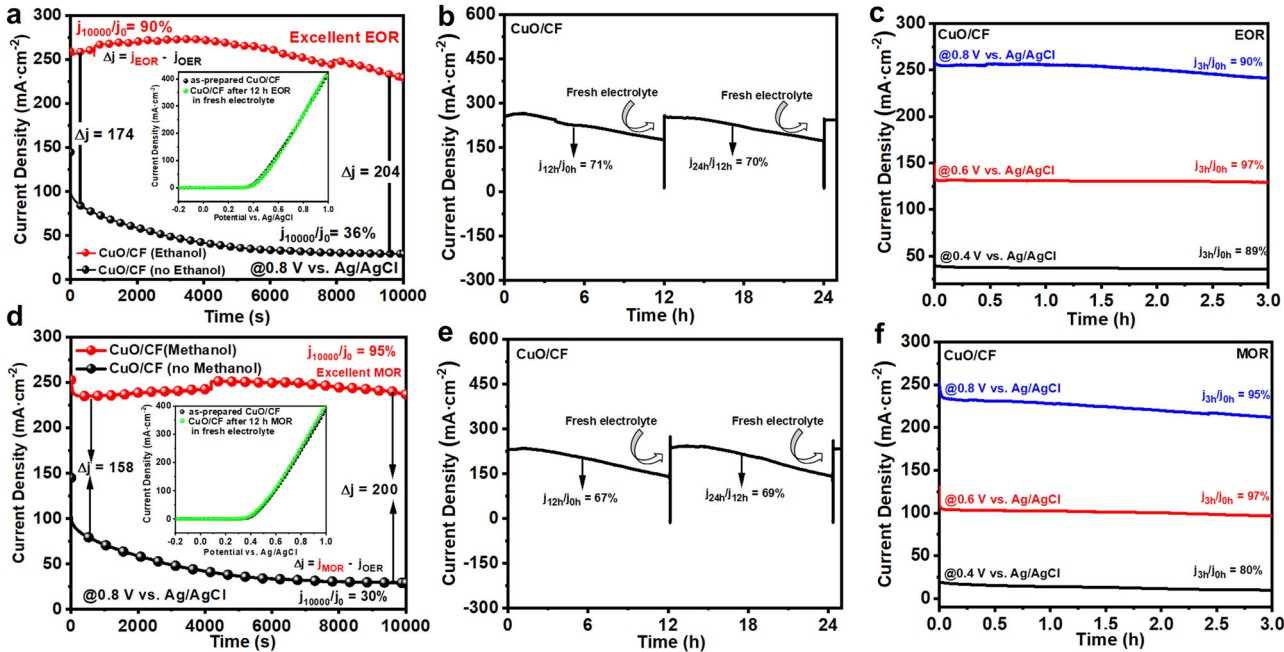

**Fig. 5 Long-term AOR stability test of the as-prepared CuO/CF. a**, **d** Chronoamperometric curves recorded in $N_2$-purged 1 M KOH with and without 1 M EtOH or MeOH at 1.82 V vs. RHE for 10000 s. **b**, **e** Chronoamperometric cyclic stability test of the as-prepared CuO/CF for two consecutive cycles at 1.82 V vs. RHE in $N_2$-purged 1 M KOH with 1 M EtOH or MeOH. **c**, **f** Chronoamperometry test for 3 h at different potentials in $N_2$-purged 1 M KOH containing 1 M EtOH or MeOH.

for further details[41]. The as-prepared CuO/CF exhibit an outstanding $TOF_{acetate}$ (5.8k $s^{-1}$) and $TOF_{formate}$ (6k $s^{-1}$) normalized by their respective FE at an applied potential of 1.82 V vs. RHE (Supplementary Fig. 15c, d). Interestingly, most of the reported EOR and MOR catalysts present an inverse behavior when it comes to their activity and selectivity towards the product at a given applied potential. For instance, most of the time the selectivity towards the product is relatively higher at lower applied potential but the corresponding current density is very low. Thus, enhancing the current density alters selectivity for the target product due to the competing side reaction like OER or the catalyst generates multiple products at the relatively higher applied potential[9,42,43]. But in our case, the abundant $V_O$ regulate the surface of the as-prepared CuO/CF, which makes them achieve higher EOR and MOR activities at low overpotential with higher selectivity for value-added products. Moreover, long-term EOR and MOR at a static potential of 1.82 V vs. RHE were carried out to measure the yield and selectivity of acetate and formate at varied periods (Supplementary Fig. 21a-d). In the case of EOR, a maximum yield of 85% was obtained after 24 h of continuous EOR with a constant selectivity of ~100% for acetate calculated after different intervals of EOR. While in the case of the MOR the obtained yield of formate was found to be 79% after 24 h of MOR and the selectivity for formate was also ~100% during the first 6 h of MOR which later dropped to 95% and then remained constant around 97% during onward MOR. It is evident from Supplementary Fig. 20b, that the $^{13}$CNMR spectra possess a miner peak related to the carbonates after 6 h of MOR, which may come from the partial oxidation of formate produced during MOR. According to the literature, the CO oxidation peak in the reverse sweep CV curves is regarded as a signature of the oxidation of adsorbed CO (an intermediate towards $CO_2$ product) produced during MOR[15,44]. But in our case, the CV curves in MOR collected over the as-prepared CuO/CF do not present such type of feature, which further endorses the selective oxidation of methanol into formate. Additionally, we studied the possible

effect of CO on the activity of the as-prepared CuO/CF, as presented in Supplementary Fig. 22a-c, the saturation of 1 M KOH containing 1 M MeOH with CO does not reduce the MOR activity of the as-prepared CuO/CF electrode, rather the as-prepared CuO/CF efficiently oxidizes the saturated CO in an electrolyte solution see Supplementary Note 8 for more details.

**Long-term AOR stability of the as-prepared CuO/CF.** The stability and durability of the electrocatalyst is crucial criteria. The as-prepared CuO/CF electrodes were assessed via chronoamperometry (CA) in 1 M KOH solution comprised of 1 M MeOH or 1 M EtOH at a potential of 1.82 vs. RHE for different durations. Fig. 5a, d represents that the current densities obtained over the as-prepared CuO/CF in EOR, and MOR were quite higher than that of OER at identical applied voltage. Initially, the difference in current densities between EOR, MOR, and OER was found to be $\Delta j_{EOR-OER} = 174$ mA·$cm^{-2}$ and $\Delta j_{MOR-OER} = 158$ mA·$cm^{-2}$, while as the experiment proceeds forward the difference in current densities increases and reaches up to $\Delta j_{EOR-OER} = 216$ mA·$cm^{-2}$ and $\Delta j_{MOR-OER} = 210$ mA·$cm^{-2}$ at 5000 s, which indicates the excellent stability of the as-prepared CuO/CF during EOR and MOR. However, a slight decline in the current densities is experienced when the EOR and MOR advance onward from 5000 s and the activity of the catalyst drops to EOR $j_{10000}/j_0 = 90\%$ and MOR $j_{10000}/j_0 = 95\%$ after 10000 s. The inset of Fig. 5a, d displays the CV curves collected over the as-prepared CuO/CF before and after long-term EOR and MOR stability tests for 10000 s in the fresh electrolyte. As evidenced the as-prepared CuO/CF after the long-term stability test regains its original activity in fresh electrolytes and exhibits a quite similar CV profile compared to the one obtained before the long-term EOR and MOR stability tests. Hence the slight decline in the activity during the long-term EOR and MOR are solely linked to the depleting concentration of ethanol and methanol[6,9,45]. The durability of the as-prepared catalyst for EOR and MOR was further evaluated with a cyclic stability test at a static potential. As presented in Fig. 5b, e, after

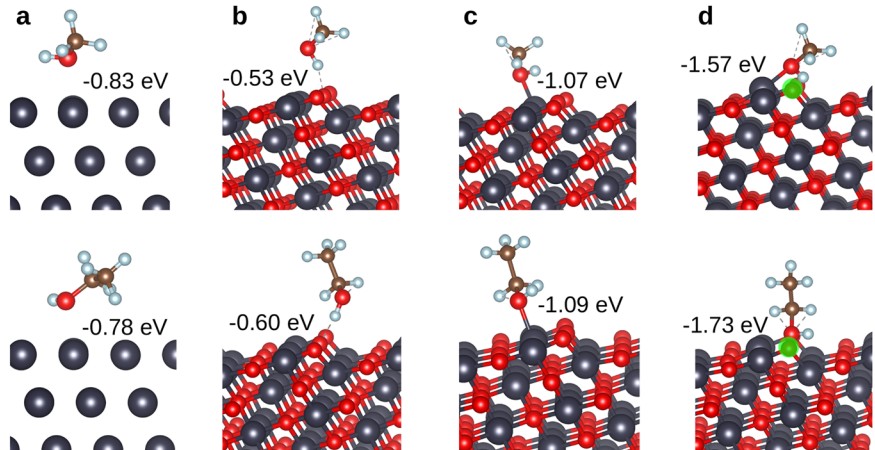

**Fig. 6 Ball-stick model for methanol and ethanol adsorption on copper and copper oxide surfaces. a** Cu(111), **b** pristine CuO(111) for interaction with the H atom, **c** pristine CuO(111) for interaction with the O atom, and **d** O-vacant CuO(111) for interaction with the O atom. The position of the $V_O$ is highlighted in green color. The binding energy is given in each case. The blue, gray, red, and black spheres represent the H, C, O, and Cu atoms, respectively.

12 h of EOR and MOR an identical portion of fresh electrolyte was added to the electrochemical cell and the experiment was run for three consecutive cycles (26 h). Interestingly, after each cycle of the stability test, the catalytic activity of the as-prepared CuO/CF enhances in fresh electrolytes. This phenomenon is quite understandable, as discussed earlier during the long-term AOR new $V_O$ as well as nanosheets are formed on the catalyst surface which simultaneously enhances the active sites and surface area of the catalyst for further AOR. Thus, it is advantageous that the as-prepared CuO/CF gets more active sites during the first cycle of the AOR and is ready to perform AOR activity more efficiently in the next cycle (fresh electrolyte). Moreover, the EIS study after the long-term EOR and MOR also reveals the reduction of $R_\Omega$ and $R_{CT}$ for the as-prepared CuO/CF (Supplementary Fig. 17a, b) which manifests that the formation of new $V_O$ and nanosheets during the long-term EOR and MOR enhances charge transfer properties between reactant molecules and catalyst. Fig. 5c, f displays the EOR and MOR stability test carried out at different applied potentials for 3 h. The as-prepared CuO/CF exhibits good stability in a broader range of applied potential. Thus, the as-prepared CuO/CF electrode is one of the best non-noble metal-based electrocatalysts which exhibits an outstanding EOR and MOR stability for more than 24 h at higher current densities of ($j_{EOR}$ = 274, $j_{MOR}$ = 247). In contrast, the previously reported non-noble metal-based electrocatalysts have been subjected to such a stability test for a very short duration of time <3000 s at a lower current density.

**DFT calculations**. To compare the methanol/ethanol adsorption activities of pristine and O-vacant CuO(111) with that of Cu(111) at the atomistic level, we perform density functional theory calculations. Ball-stick models and the obtained binding energies are shown in Fig. 6. The methanol/ethanol molecules can bond with Cu(111) and O-vacant CuO(111) only by their O atoms whereas they can bond with pristine CuO(111) by their H atoms (with O atoms) or by their O atoms (with Cu atoms). We find that bonding by the O atoms is energetically highly favorable. The bonding is much stronger on pristine CuO(111) than on Cu(111). It is much stronger on O-vacant than pristine CuO(111) due to enhanced charge transfer between the molecules and substrate. The enhanced bonding with O-vacant CuO(111) is consistent with our experimental results.

## Conclusion
In conclusion, we report the in-situ fabricated $V_O$-rich CuO/CF nanosheet as a monolith electrode via a simple ultrasonication-assisted acid-etched method. The as-prepared CuO/CF acts as an outstanding platform for the electrooxidation of alcohols in alkaline media. The obtained $V_O$-rich ultrathin CuO/CF nanosheets exhibit an outstanding activity for the selective oxidation of ethanol and methanol into value-added acetate and formate with a reasonably higher Faraday efficiencies ~99% at an applied potential of 1.82 V vs. RHE. DFT calculation reveals that $V_O$ enhances charge transfer and efficiently tunes the electronic structure of the surface, facilitating the adsorption of reactant molecules on the surface of as-prepared CuO/CF, which results in exceptionally boosted MOR and EOR kinetics. The outstanding CO tolerance ability as well as the in-situ grown active sites bestow the as-prepared CuO/CF with significantly enhanced stability for an exceptionally long time. So, this study not only signifies the vital role of $V_O$ in enhancing the AOR kinetics at the atomic level but also encourages exploring other transition metal-based catalysts for stable and selective oxidation of alcohols. Meanwhile, adding ultrasound energy with acid-etching is a simple and promising path for the large-scale fabrication of other transition metal-based oxide/hydroxide catalysts for AOR.

## Methods
**Reagents and materials**. Methanol, ethanol, and potassium hydroxide were acquired from Aldrich and hydrochloric acid (HCl, 37%) was obtained from Sinopharm. All chemicals were used without further purification. CF with an areal density of 550 g/m² and a pore size of 130 PPI and thickness of 1.0 mm were purchased from Cyber Electrochemical Material. In all our experiments, deionized water (18.6 MΩ cm), acquired from a Mili-QWater purification system was used.

**Preparation of $V_O$-rich CuO/CF**. The $V_O$-rich CuO/CF monolith electrode was fabricated by a simple ultrasonication-assisted acid-etched method. In a typical procedure, the surface of B-CF (1 × 1 cm) was oxidized with the help of ultrasound in a 2 M HCl solution for 10 min. This was followed by washing and drying steps in the open air at a relatively higher pH and then collected for further characterization.

**Preparation of B-CF**. A clean piece of CF (1 × 1 cm) was thoroughly washed with water and ethanol and then subjected to drying in the open air for 30 min. After the drying step, the obtained B-CF was collected for further characterization.

**Materials characterization**. A field emission scanning electron microscope (ZEISS GeminiSEM 500) was used to obtain SEM images, while TEM images were acquired with a field emission transmission electron microscope (JEM-2011F).

X-ray diffractometry experiments of the samples were performed with the Rigaku-D X-ray diffractometer using Cu Kα ($\lambda = 1.54178$ Å) radiation. XPS of the samples were obtained with Thermos ESCALAB 250 employing Al Kα ($\gamma = 1486.6$ eV) radiation source for the excitation. EDX spectra of the samples were acquired with Aztec X-Max80 equipped with ZEISS GeminiSEM 500. The LABRAM-HR Raman system was used to record Raman spectra. All samples were dried before characterization.

**Electrochemical methods**. The electrochemical workstation CHI 660E with a three-electrode system was used to perform the electrochemical test in 1 M KOH. A plate of Pt (10 cm$^2$) was used as a counter electrode while an Ag/AgCl (3 M KCl) and Hg/HgO were used as reference electrodes (see Supplementary Note 2 and Supplementary Fig. 4a–d for details). CV was conducted at a scan rate of 5 mV·s$^{-1}$ in N$_2$-purged 1 M KOH with and without 1 M alcohols. SCV curves were obtained from the CA response of as-prepared CuO/CF at constant potential ranging from 1.023 to 2.023 V vs RHE in N$_2$-purged 1 M KOH with and without 1 M ethanol or 1 M methanol. Long-term durability tests were conducted with CA at different applied potentials in 1 M KOH either containing 1 M ethanol or 1 M methanol. The EIS measurements were performed at 0.45 V vs. Ag/AgCl at an amplitude of 5 mV in a frequency range of 100 kHz to 0.05 Hz. Backward CV-sweep collected at 200 mV·s$^{-1}$ were used to calculate electrochemically accessible active sites ECAS for each catalyst via charge integration method.

**Computational methods**. Spin-degenerate density functional theory calculations were performed using the Vienna ab initio simulation package with the projector-augmented wave method[46]. The electronic exchange-correlation potential was modeled in the generalized gradient approximation of Perdew-Burke-Ernzerhof. The van der Waals interaction was described by the zero-damping method of Grimme[47]. The bulk structures of Cu and CuO were relaxed and Cu(111) and CuO(111) surfaces were formed with a vacuum layer of 15 Å thickness in the out-of-plane direction. $2 \times 2 \times 1$ supercells were built to guarantee a distance of at least 10 Å between the adsorbed methanol/ethanol molecules. One of the O atoms was removed from the CuO(111) surface to study the effect of O deficiency. The atoms in the two bottom layers of the substrates were fixed while the others were relaxed. The Brillouin zone was sampled on $7 \times 7 \times 1$ Γ-centered $k$-meshes. A cut-off energy of 500 eV was used for the plane-wave basis. The Hellmann-Feynman forces were converged to $10^{-3}$ eV/Å and the total energies were converged to $10^{-7}$ eV. The methanol/ethanol binding energy was defined as $E_b = E_{substrate+methanol/ethanol} - E_{methanol/ethanol} - E_{substrate}$, where $E_{substrate+methanol/ethanol}$, $E_{methanol/ethanol}$, and $E_{substrate}$ are the total energies of the combined system, molecule, and substrate, respectively.

## Data availability
The authors declare that all data supporting the findings of this study are available within the article and its supplementary information files, and from the corresponding author on reasonable request.

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

## Acknowledgements
This work was supported by funding from the Natural Science Foundation of Anhui Province (1908085J19), the National Natural Science Foundation of China (21722406, 21975240), and King Abdullah University of Science and Technology (KAUST). Mustafa Khan acknowledges the Chinese Academy of Science (CAS) and TWAS for supporting his Ph.D. degree from the University of Science and Technology of China in the category of the 2019 CAS-TWAS President's Fellowship Awardee (series no. 2019–184).

Muhammad Imran Abdullah acknowledges the local challenge fund "LCF-7 (2020)" World Bank-funded HEDP project by the Higher Education Commission (HEC) Pakistan for supporting his postdoctoral research from Government College University Lahore Pakistan.

## Author contributions
U.S. and M.M. supervised the project. M.K. and A.H. contributed to the experiments, data analysis and first draft of the manuscript. A.S. performed the density functional theory calculations. M.I.A., R.S.A. and B.G.P. assisted in the electrochemical experiments and analysis. T.M., A.A. and N.Z. conducted the catalyst characterization and NMR analysis. All authors discussed the results and assisted in the preparation of the manuscript.

## Competing interests
The authors declare no competing interests.
