## [Peer Review File · Communications Chemistry]

Reviewers' comments:

Reviewer #1 (Remarks to the Author):

This is an interesting paper that presents the work with " Ultrasonication-assisted Preparation of Oxygen-vacancy rich CuO/CF Self-supported Electrode: Boosting Selective Oxidation of Alcohols to Value-added Chemicals ". In this work, the author reported the in-situ fabrication of oxygen vacancies-rich CuO nanosheets on copper foam (CF) via a simple ultrasonication-assisted acid etched method. The samples have been well characterized and the mechanism has been clearly clarified. The paper is recommended to be accepted after addressing the following issues:

1. The introduction part should be improved: I) The background for choosing transition metal-based electrocatalysts for AOR should be discussed. II) There have been many works about preparing Cu-based catalysts for AOR reduction, so the difference between this work and previous works should be clearly expounded.
2. The obtained thickness of nanosheets is ca. 15.6 nm, so it is not suitable to describe them as "ultrathin" nanosheets.
3. It would be better to use TEM and element mapping to further analyze the structure and composition of the samples.
4. The related materials characterizations after stability test need to be conducted and discussed, including XRD, SEM and TEM etc.
5. Some references related with surface defects are suggested to be cited, such as *Joule* 2018, 12, 2551-2582; *Adv. Mater.* 2021, 2106354; *Angew. Chem. Int. Ed.* 2021, 60, 7602 -7606; *Rare Met.* 2021 40, 3019-3037

Reviewer #2 (Remarks to the Author):

This work presents an interesting way of making CuO nanostructures on Cu foam using ultrasonication and its eventual use in MeOH and EtOH oxidation to their respective carboxylates. Though the work is intriguing, there are several issues that need to be addressed before it can be recommended for publication.

- 1) Authors stated that MOR and EtOR catalysts can be used in DAAFCs. This is not entirely correct particularly when the overpotentials of MOR and EtOR are larger than the overpotentials of ORR in an alkaline medium. In such a case, it is literally impossible to construct a fuel cell. However, a hybrid system may work.
- 2) The superiority of this method of oxidizing Cu to CuO over other faster methods such as chemical oxidation and electrochemical anodization (e.g. *ACS Appl. Energy Mater.* 2021, 4, 1, 899-912 and *ACS Appl. Energy Mater.* 2022, 5, 1, 419-429.) is neither poorly justified nor justified at all.
- 3) Use of Ag/AgCl in alkaline is a strongly questioned practice. Hence, I advise the authors to acquire the data with Hg/HgO instead. Using a reference electrode with an inner solution of pH ~7 in an electrolyte of pH 14 basically forms a concentration cell. This can cause some serious issues in determining activity. The difficulties worsen as the time of exposure of Ag/AgCl to 1.0 M KOH increases.
- 4) The apparent activity markers reported are truly of practical importance but one should not overlook the intrinsic markers too for these are what lead to the construction of efficient catalytic electrodes with high apparent activities. Hence, the authors must properly determine TOF and specific activity. Please see <https://doi.org/10.1002/anie.202110352> and <https://doi.org/10.1021/acsaem.0c02822>

Besides, transient CV and LSV are always exaggerating the activity of an electrocatalyst and hence, an appropriate alternative technique should be used to determine the real activity free from double-layer charging currents. Please see Sengeni Anantharaj et al 2022 J. Electrochem. Soc. 169 014508.

5) I could also note that the CV data points were used for Tafel analysis but no information on iR drop correction is provided. Moreover, even with 100% iR drop correction, using CV/LSV responses could never serve the purpose of accurate Tafel analysis. Please see ACS Energy Lett. 2021, 6, 4, 1607–1611.

Reviewers' comments:

Manuscript ID: COMMSCHEM-22-0093-T

Ultrasonication-assisted Preparation of Oxygen-vacancy-rich CuO/CF Monolith Electrode: Boosting Selective Oxidation of Alcohols to Value-added Chemicals

Authors: Mustafa Khan¹, Asima Hameed^{1,6}, Abdus Samad², Talifhani Mushiana¹, Muhammad Imran Abdullah^{1,5*}, Asma Akhtar¹, Raja Shahid Ashraf⁵, Ning Zhang³, Bruno G. Pollet⁴, Udo Schwingenschlögl^{2*}, Mingming Ma^{1*}

To all reviewers:

We would like to replace the terminology “Self-supported” with “Monolith” in the title of our manuscript since it emphasizes better our aim and the approach, we used to address it in the manuscript.

Response to Reviewer 1.

Comments and suggestions from Reviewer 1.

This is an interesting paper that presents the work with " Ultrasonication-assisted Preparation of Oxygen-vacancy rich CuO/CF Self-supported Electrode: Boosting Selective Oxidation of Alcohols to Value-added Chemicals ". In this work, the author reported the in-situ fabrication of oxygen vacancies-rich CuO nanosheets on copper foam (CF) via a simple ultrasonication-assisted acid etched method. The samples have been well characterized and the mechanism has been clearly clarified. The paper is recommended to be accepted after addressing the following issues:

Response: Thank you very much for the positive comment and recommendation. Please find the following detailed responses to your comments and suggestions.

1. The introduction part should be improved: I) The background for choosing transition metal-based electrocatalysts for AOR should be discussed. II) There have been many works about preparing Cu-based catalysts for AOR reduction, so the difference between this work and previous works should be clearly expounded.

Response: We have improved the introduction part accordingly. I) Please find a comprehensive background discussion regarding the use of transition metal-base electrocatalyst for AOR. Alcohol oxidation reaction at the anode is the key half-reaction that determines the overall efficiency of an alkaline electrolyzer. Currently, AOR is thoroughly carried out with expensive Pt and their alloys with other noble metals like Pt-Ru and Pt-Pd, etc. The noble-metal-based catalysts completely oxidize the alcohols into CO₂ and the generated CO intermediate will poison the catalyst, therefore the high price as well as vulnerable nature of these catalysts towards CO poisoning restrict their use for AOR. On the other hand, low-cost earth-abundant transition metal-based catalysts selectively oxidize alcohols into carboxylates as value-added chemicals. The latter path is economically and environmentally advantageous in terms of lowering the overall cost of the process as well as avoiding CO₂ emissions into the environment. These information has been added to the revised manuscript (Page 2).

II) Please find the superiority of our method over other related methods. Easy and fast fabrication of an efficient catalyst is the bottom line for the successful commercialization of any specific application. Therefore, researchers are focusing on faster and simple methods for the fabrication of an efficient catalysts. Recently, Ananthaj *et al.* have reported a faster anodization method that takes only 80 seconds to fabricate a dense array of Cu(OH)₂-CuO nanoneedles on Cu foil substrate *via* reasonably lower constant potential in 1M KOH. They further modified the anodization method by introducing Cu foam substrate containing Ni impurities to grow Cu-O/(Ni) nanowires, which takes 10 mins. The Ni impurity in Cu-O/(Ni) nanowires significantly enhances the MOR activity as compared to the Cu(OH)₂-CuO nanoneedle arrays. The catalysts prepared with the anodization method developed by Ananthaj *et al.* take sufficiently lower time and have better MOR performance as compared to that of Cu-based catalysts reported earlier by electrochemical as well as other physical and chemical methods. We have prepared in

situ CuO nanostructure on copper foam (CF) *via* a simple and mild chemical oxidation method. Unlike other chemical oxidation methods which usually take longer time and require different chemical oxidants under strict reaction conditions. Our method uses O₂ in the air as an oxidant under ambient conditions. Our developed method also takes only 10 mins for the fabrication of CuO nanostructure like that of the reported faster anodization method discussed earlier. Moreover, the developed chemical oxidation method also imparts some defects in the form of oxygen vacancies (Vo) into the CuO nanostructure which are known to improve the catalytic performances of the electrocatalyst. As a result, the obtained CuO catalyst show superior MOR and EOR performance specifically in terms of intrinsic properties like specific activity and exceptional turnover frequency (TOF) for value-added formate and acetate compared to the Cu-based oxides/hydroxides as well as most of the oxides/hydroxides reported for other transition metals. These information has been added to the revised manuscript (Page 3). Manuscript amended.

2. The obtained thickness of nanosheets is ca. 15.6 nm, so it is not suitable to describe them as “ultrathin” nanosheets.

Response: We agree to your suggestion and have replaced the “ultrathin” nanosheets terminology with nanosheets throughout the revised manuscript.

3. It would be better to use TEM and element mapping to further analyze the structure and composition of the samples.

Response: We admire your suggestion to further analyze the structure and composition of our samples with TEM and elemental mapping. Please find the detailed TEM and elemental mapping of our samples in Figure 3 of the manuscript. The TEM analysis further confirms the results obtained with SEM analysis as well as the elemental mapping confirm the uniform distribution of Cu and O throughout the synthesized nanostructure. These information has been added to the revised manuscript (Page 4, 5). Manuscript amended.

Revised Figure 2. SEM images of, a) B-CF, b) UAAE-CF before the AOR stability test, c) UAAE-CF after the AOR stability test. TEM images of, d) UAAE-CF before the AOR stability test, e, f) UAAE-CF after the AOR stability test. g) Raman spectra of B-CF and UAAE-CF before and after the AOR stability test. h) XRD patterns UAAE-CF, i) XRD patterns of UAAE-CF after the AOR stability test. Elemental mapping of, j) B-CF, k) UAAE-CF, l) UAAE-CF after the AOR stability test.

4. The related materials characterizations after stability test need to be conducted and discussed, including XRD, SEM and TEM etc.

Response: We appreciate your comment regarding the characterization of the material after the stability test. We have thoroughly characterized the catalyst material after the stability test with related characterization techniques including XRD, XPS, SEM, TEM, and Raman analysis. See Fig. 2 of the revised manuscript. These characterization techniques have been discussed after the stability test in the revised version of our manuscript (page 4, 5, 6). Manuscript amended.

5. Some references related with surface defects are suggested to be cited, such as Joule 2018, 12, 2551-2582; Adv. Mater. 2021, 2106354; Angew. Chem. Int. Ed. 2021, 60, 7602–7606; Rare Met. 2021 40, 3019–3037.

Response: Thank you for suggesting these valuable references. These references are critically important and closely related to this manuscript, therefore have been cited in the revised manuscript (Ref. 26, 27, 28 and 29).

Response to Reviewer 2.

Comments and suggestions from Reviewer 2.

This work presents an interesting way of making CuO nanostructures on Cu foam using ultrasonication and its eventual use in MeOH and EtOH oxidation to their respective carboxylates. Though the work is intriguing, there are several issues that need to be addressed before it can be recommended for publication.

Response: Thank you very much for your positive comments and recommendations. Please find the following detailed responses to your comments and suggestions.

1) Authors stated that MOR and EtOR catalysts can be used in DAAFCs. This is not entirely correct particularly when the overpotentials of MOR and EtOR are larger than the overpotentials of ORR in an alkaline medium. In such a case, it is literally impossible to construct a fuel cell. However, a hybrid system may work.

Response: We agree to your suggestion, practically it is not possible to construct DAAFCs when the MOR and EtOR overpotentials of a catalyst are larger than the ORR overpotential in an alkaline electrolyte. However, our MOR and EtOR catalysts can be used more efficiently to replace the energy-intensive oxygen evolution reaction in a different hybrid electrolytic cell. The selective oxidation of alcohols to their respective carboxylates offers a significant reduction in energy input as well as generates value-added chemicals and fuel which is economically and environmentally more desirable than their full oxidation to CO₂. We have updated this information in the abstract and introduction of the revised version of our manuscript (page 1 and 2). Manuscript amended.

2) The superiority of this method of oxidizing Cu to CuO over other faster methods such as chemical oxidation and electrochemical anodization (e.g. ACS Appl. Energy Mater.

2021, 4, 1, 899–912 and ACS Appl. Energy Mater. 2022, 5, 1, 419–429.) is neither poorly justified nor justified at all.

Response: Please find the superiority of our method over the advised and other related methods. Easy and fast fabrication of an efficient catalyst is the bottom line for the successful commercialization of any specific application. Therefore, researchers are focusing on faster and simple methods for the fabrication of an efficient catalysts. Recently, Ananthaj *et al.* have reported a faster anodization method that takes only 80 seconds to fabricate a dense array of Cu(OH)₂-CuO nanoneedles on Cu foil substrate *via* reasonably lower constant potential in 1M KOH. They further modified the anodization method by introducing Cu foam substrate containing Ni impurities to grow Cu-O/(Ni) nanowires, which takes 10 mins. The Ni impurity in Cu-O/(Ni) nanowires significantly enhances the MOR activity compared to the Cu(OH)₂-CuO nanoneedle arrays. The catalysts prepared with the anodization method developed by Ananthaj *et al.* takes sufficiently lower time and has a better MOR performance compared to the Cu-based catalysts reported earlier by electrochemical as well as other physical and chemical methods. We have prepared in situ CuO nanostructure on copper foam (CF) *via* a simple and mild chemical oxidation method. Unlike other chemical oxidation methods which usually takes longer time and require different chemical oxidants under strict reaction conditions, our method uses O₂ in the air as an oxidant at ambient conditions. Our developed method also takes only 10 mins for the fabrication of CuO nanostructure like that of the reported faster anodization method discussed earlier. Moreover, the developed chemical oxidation method also imparts some defects in the form of oxygen vacancies (Vo) into the CuO nanostructure which are known to improve the catalytic performances of the electrocatalyst. As a result, the obtained CuO catalyst show superior MOR and EOR performance specifically in terms of intrinsic properties like specific activity and exceptional turnover frequency (TOF) for value-added formate and acetate compared to the Cu-based oxides/hydroxides as well as most of the oxides/hydroxides reported for other transition metals. These information has been added to the revised manuscript (Page 3). Manuscript amended.

3) Use of Ag/AgCl in alkaline is a strongly questioned practice. Hence, I advise the authors to acquire the data with Hg/HgO instead. Using a reference electrode with an inner solution of pH ~7 in an electrolyte of pH 14 basically forms a concentration cell. This can cause some serious issues in determining activity. The difficulties worsen as the time of exposure of Ag/AgCl to 1.0 M KOH increases.

Response: We highly appreciate your concerns about the use of Ag/AgCl as a reference electrode in alkaline electrolyte. As it is very much possible to obtain a misleading electrocatalytic activity while using Ag/AgCl as a reference electrode in alkaline electrolyte. For this purpose, we compare the EOR electrocatalytic activity for the same electrocatalyst in the same electrolyte with Ag/AgCl and Hg/HgO as reference electrodes. Figure S4a, shows the EOR activity of the as-prepared CuO/CF measured in the same electrolyte solution with Ag/AgCl and Hg/HgO as reference electrodes in a potential window ranging from -0.2 to 1 V vs Ag/AgCl or Hg/HgO. Figure S4b, displays the EOR data in the reversible hydrogen electrode (RHE) scale obtained with Ag/AgCl and Hg/HgO reference electrodes. As presented the EOR activity obtained for the as-prepared CuO/CF with both Ag/AgCl and Hg/HgO reference electrodes were quite similar. Furthermore, a chronoamperometry (CA) experiment was carried out at an applied potential of 0.8 V vs Ag/AgCl for the duration of 11 hours to notice if any possible change in the EOR activity for the as-prepared CuO/CF after the prolonged use of Ag/AgCl in CA test (Figure S4c). As displayed in Fig. S4d, the LSV curves taken before and after CA test were also quite similar which nullifies the possibility of reporting misleading activity in this work. However, to be on the safe side we will strictly follow the use of Hg/HgO as a reference electrode in the alkaline electrolyte in our further projects. These information has been added to the revised supporting information (Page S5 and S6). Supporting information amended.

Figure S4. Comparative EOR activity of the as-prepared CuO/CF obtained with different reference electrodes. a) EOR activity of the as-prepared CuO/CF obtained with Ag/AgCl and Hg/HgO as reference electrodes. b) EOR activity of the as-prepared CuO/CF obtained with Ag/AgCl and Hg/HgO in Reversible Hydrogen Electrode (RHE) scale. c) Long-term stability test conducted at 0.8 V vs Ag/AgCl for 11 hours. d) Comparative LSVs before and after long-term stability test for the as-prepared CuO/CF using Ag/AgCl as reference electrode after being used for 11 hours in stability test.

4) The apparent activity markers reported are truly of practical importance but one should not overlook the intrinsic markers too for these are what lead to the construction of efficient catalytic electrodes with high apparent activities. Hence, the authors must properly determine TOF and specific activity. Please see <https://doi.org/10.1002/anie.202110352> and

<https://doi.org/10.1021/acsaem.0c02822> Besides, transient CV and LSV are always exaggerating the activity of an electrocatalyst and hence, an appropriate alternative technique should be used to determine the real activity free from double-layer charging currents. Please see Sengen Anantharaj et al 2022 J. Electrochem. Soc. 169 014508.

Response: We highly appreciate your insightful suggestion on the importance of reporting the intrinsic markers like TOF and specific activity which determine the superiority of the catalytic electrode in question. We find the suggested references critically important for the accurate calculation of TOF and specific activity and follow these references to report TOF and specific activity. See Fig. S15a-b for calculated TOF and specific activity and related discussion in the revised manuscript (page 10 and 11). Besides transit CV and LSV we have determined the real AOR activity of the catalyst-free from double-layer charging current *via* sample current voltammetry (SCV) following the suggested references. Find the AOR activity measured with SCV and their comparison with AOR activity measured with LSV in Fig. 4b and Fig. S6a-f as well as the related discussion in the revised manuscript (page 8 and 9). Manuscript and supplementary information amended.

Figure S15. Comparison of specific activity for B-CF and as-prepared CuO/CF. a) MOR current densities normalized by the relative ECAS. b) EOR current densities normalized by the relative ECAS. Turnover Frequency (*TOF*) of as-prepared CuO/CF for acetate and formate at given applied potential normalized by their respective Faraday efficiencies. c) $TOF_{Acetate}$ and d) $TOF_{Formate}$.

Figure S6. Comparative MOR and EOR activity of the as-prepared CuO/CF measured with LSV and SCV. a,d) MOR and EOR data for the as-prepared CuO/CF obtained with LSV (Black dots) and SCV (Red dots). b,e) Bar graphs for MOR and EOR activity obtained with LSV and SCV compared at different potentials. c,f) Summaries of (%) difference in MOR and EOR current densities measured with LSV and SCV at different potentials.

5) I could also note that the CV data points were used for Tafel analysis but no information on *iR* drop correction is provided. Moreover, even with 100% *iR* drop correction, using CV/LSV responses could never serve the purpose of accurate Tafel analysis. Please see ACS Energy Lett. 2021, 6, 4, 1607–1611.

Response: Thank you very much for the positive comment and recommendation. We have derived the Tafel plots from the respective SCV data with 100% *iR* correction

according to the suggested reference. See Fig. 4e and related discussion on Tafel analysis in the revised manuscript (page 9). Manuscript amended.

Revised Figure 4. AOR performance of the as-prepared CuO/CF. a) CV curves collected over the as-prepared CuO/CF in nitrogen-purged 1M KOH with and without 1M ethanol or methanol. b) Sample Current Voltammetry (SCV) curves collected over the as-prepared CuO/CF in nitrogen-purged 1M KOH with and without 1M ethanol or methanol. c) Summaries of anodic potentials at 50, 100, 150, and 200 mA·cm⁻² for the as-prepared CuO/CF in 1M KOH with and without 1M ethanol (EtOH) or methanol (MeOH) based on their respective SCV data. d) Comparison of EOR, MOR, and OER Tafel plots obtained from their respective SCVs collected over the as-prepared CuO/CF with 100% iR correction. e) Comparison with recently reported non-noble metal-based MOR and EOR catalysts, star data represent activity of the as-prepared CuO/CF (see Table S2,3 of supporting information for more details). f) Summaries of Faraday's efficiencies for acetate and formate.

REVIEWERS' COMMENTS:

Reviewer #1 (Remarks to the Author):

It can be accepted at this state.

Reviewer #2 (Remarks to the Author):

The authors have made satisfactory revisions and it may now be accepted.